# OpenReview forum: "Principle-Driven Self-Alignment of Language Models from Scratch with Minimal Human Supervision"
_NeurIPS.cc/2023/Conference — NeurIPS 2023 spotlight_

### Official Review · Reviewer_PEmx · 2023-07-04

**Soundness:** 3 good
**Presentation:** 3 good
**Contribution:** 3 good
**Rating:** 7
**Confidence:** 4

**Summary:**

The paper proposes a new method to generate diversified, principle-guided synthetic data from LLM itself to ease requirement of large amount of annotated instruction-following data for supervised fine-tuning task for LLMs. The generation process follows four steps. First step is to generate adversarial topic-guided instructions, second step is generate responses using in-context learning with pre-defined principles, third step is supervised fine-tuning with generated instruction-following data. Fourth step is to leverage context distillation for verbose cloning. `LLaMA-65b` model is fine-tuned with the data generated from the aforementioned approach. Evaluations are conducted on TruthfulQA and Bigbench-HHH Eval dataset. The fine-tuned model shows strong performance.


**Strengths:**

The paper combines the approaches of self-instruct paper and constitution AI paper and proposed the topic-guided, principle-following self-instruct way of data generation which could ease requirement of large amount of annotated data and improve diversity of instruction-following data. Although nothing is new, the paper follows a clear and logical path to generate synthetic data in the era of LLMs. Given a strong pretrained model, we have reason to believe this will work pretty well which is to some extent verified in the evaluation results.

**Weaknesses:**

- The pretrained LLMs are not instruction fine-tuned. It could be challenging to generate clean topic-guided instructions and principle-following response. The paper didn't talk if there is any filtering step following these generation steps.
- How much data is generated, how much data are there in the eval set. These numbers are not shown in the paper clearly.
- Evaluation is not enough. We might need more evaluations on instruction-following dataset
- Verbose clone is not working well, and no reason or discussion is given as why this is the case.

**Questions:**

- The topics are generated from a red-teaming (adversarial) way, where the LLM is not supposed to perform well. Then how do we guarantee the quality of generation of responses?
- Can you give more intuition of verbose tax in the paper instead of just giving the concept? The `alignment tax` is only seen on smaller scale models.
- Can you tell how much data are generated in the process and used to fine-tune the model
- At the starting point, when model is not trained to understand instructions, can it generate topics and new instructions well?



**Limitations:**

None is given

---

> ### Author Rebuttal · Authors · 2023-08-10
>
> We sincerely thank you for the thorough review and positive feedback on our work. Your questions/comments/suggestions are invaluable for the improvement of our method and the revision of the manuscript. Let us address each individual comment/question below.
>
> > Concerning the quality of response generation for red-teaming prompts:
>
> We've designed principles including ethical, question assessment, candor, dated knowledge, static, etc., as the guidelines for the model to follow, which are trained with red-teaming and other prompts. It's worth noting that if we only train our model without red-teaming prompts, then that would be an issue, because the anti-red-teaming principles will not be triggered. However, this is not the case.
>
> > On providing a deeper understanding of the "Verbose Tax":
>
> Certainly! The concept of "Verbose Tax", as mentioned on line 287, hints at an inherent balance that the model tries to maintain between offering extensive (i.e., verbose), helpful responses and ensuring the safety and coherence of those responses [1].
>
> > Regarding the volume of data generated and utilized for fine-tuning:
>
> Detailed specifications related to the generated data's volume and its usage in the fine-tuning process are elaborated in Appendix D.
>
> > On the model's initial capability to generate topics and understand instructions:
>
> In the literature, base (unaligned) large language models have demonstrated the ability to interpret instructions and generate relevant topics [2-3]. This proficiency can be attributed to the expansive data they've been trained on, coupled with their in-context learning capability when suitably prompted [4]. Our approach inherent such a capability.
>
> ---
>
> [1] Llama 2: Open Foundation and Fine-Tuned Chat Models
>
> [2] Language Models are Few-Shot Learners
>
> [3] The Capacity for Moral Self-Correction in Large Language Models
>
> [4] Self-Instruct: Aligning Language Models with Self-Generated Instructions

---

### Official Review · Reviewer_FPHi · 2023-07-06

**Soundness:** 3 good
**Presentation:** 3 good
**Contribution:** 2 fair
**Rating:** 6
**Confidence:** 4

**Summary:**

This paper proposes Self-Align, a method for aligning a language model from scratch (without previous RLHF training) with fewer annotations. Self-Align works by using the LM to generate a set of example instructions/tasks, generating from the LM conditioned on the instruction + a human-written set of principles, and then distilling this back into the model by finetuning without the principles and demonstrations in context. The paper shows that this method improves performance/accuracy and (synthetic human) ratings on variety of benchmarks.


**Strengths:**

- simple and clearly presented method
- strong performance / thorough comparison to both open- and closed-source baselines


**Weaknesses:**

- generally seems to underperform Vicuna, even though the Dromedary model is 5x larger. I still see that the Self-Align methodology could be a contribution (with better understanding of how much the different parts of the methodology matter), but there may not be a reason to build on the Dromedary model when Vicuna is available / more accessible in size.
     - If this is the case then perhaps the paper could be stronger if it shows that the Self-Align methodology still provides improvements *on top of* Vicuna's alignment (i.e. applying Self-Align with Vicuna as the base model).
- lacking in ablations: I don't have a great sense of how much the different design choices are contributing to the final performance of the model, e.g.: sensitivity to self-instruct instructions, particularly the 20 "topic-specific prompts" (were these hand-crafted to match the downstream tasks?), importance of / sensitivity to ICL examples during self-alignment (both the specific 5 examples and the number of examples). It would be great to get a sense of e.g. variance across prompts to understand how much of the method works because of the specific prompts the authors crafted, as opposed to the methodology in general.


**Questions:**

- l36: what are "topic-specific prompts"?
- sec 3.4 verbose cloning: are the model's brief responses primarily because the ICL examples are short?
- is the verbose cloning process simply repeating principle engraving with a longer demonstration in context, or is there something else going on here?
- any idea why verbose cloning often decreases performance? Can readers conclude that verbose cloning hurts accuracy but makes the model's answers more "stylistically" aligned (hence the improvement on the Vicuna evals)?
- sec 4.2.1 truthfulQA: does Dromedary outperform Vicuna on MC? Why weren't all the models tested on MC?


**Limitations:**

Yes

---

> ### Author Rebuttal · Authors · 2023-08-10
>
> We sincerely thank you for the thorough review and positive feedback on our work. Your questions/comments/suggestions are invaluable for the improvement of our method and the revision of the manuscript. Let us address each individual comment/question below.
>
> > Regarding the difference between Dromedary over Vicuna
>
> Firstly, Vicuna is designed for knowledge distillation from ChatGPT, and constrained by non-commercial usage, Dromedary, on the other hand, allows flexibility for application to commercial models like LLaMA-2, when the Self-Align methodology is integrated. Secondly, Vicuna, being dependent on the pre-aligned ChatGPT model, cannot surpass the performance of ChatGPT. In contrast, the Self-Align approach aims to and successfully achieved the goal of enhancing the base LM model with minimal human supervision, as we have proven in the paper.
>
> > On enhancing the alignment of models like Vicuna using Self-Align:
>
> This suggestion is indeed valuable. We recognize that a model adept at instruction-following could boost the in-context learning and reasoning by the base LM, potentially rendering it a more fitting candidate for the Self-Align pipeline. However, the central goal of our approach is to minimize human supervision, and to build upon an already aligned model which does not deviate from that goal (such as those demanding massive human annotations).
>
> > Concerning the dependency of our methodology on the specificity of prompts:
>
> We understand and appreciate your concern. The intrinsic relationship between the Self-Align methodology and the specific prompts we utilize makes it a considerable challenge to conduct an ablation study with varied prompts. However, with the advent of LLaMA-2 and its extended 4k context length, we've experimented with responses modeled in the general-specific-general style in ICL examples. These have shown that Dromedary-2 achieves superior outcomes, even when negating the need for verbose cloning and inference-time few-shot instances. The results on Vicuna benchmark questions are shown below:
>
> | v.s. Vicuna-13b |  | W | T | L |
> |---|---|---|---|---|
> | Dromedary-1-65b (final) | 0-shot | 16 | 1 | 63 |
> | Dromedary-1-65b (final) | 2-shot | 19 | 8 | 53 |
> | Dromedary-2-70b (self-align-only) | 0-shot | 28 | 32 | 20 |
> | Dromedary-2-70b (self-align-only) | 2-shot | 52 | 17 | 11 |
>
> We will incorporate these findings in our revised paper.
>
> ### Clarifications
>
> 1. "Topic-specific prompts" refer to the 20 Seed Prompts designed for Topic-Guided Red-Teaming Self-Instruct, as detailed in Appendix M.
>
> 2. The succinctness of the model's responses is indeed influenced by the brevity of the ICL examples.
>
> 3. For the verbose cloning process, we leverage a distinct prompt, detailed in Appendix K, guiding the initially aligned model to produce more extensive responses.
>
> 4. The observed "Verbose Tax" (mentioned on line 287) could be attributed to an inherent balance between the model's helpfulness and safety, as delineated in [1].
>
> 4. Our decision to not test all models on MC was driven by our focus on presenting results for the base and aligned language models via RLHF and Self-Align. To maintain clarity and succinctness, we've chosen not to include results for models like Alpaca or Vicuna, which are derivatives of Text-Davinci-003 and ChatGPT.
>
> ---
>
> [1] Llama 2: Open Foundation and Fine-Tuned Chat Models

---

### Official Review · Reviewer_Yvp3 · 2023-07-07

**Soundness:** 4 excellent
**Presentation:** 4 excellent
**Contribution:** 4 excellent
**Rating:** 7
**Confidence:** 5

**Summary:**

The authors use the self-instruct approach combined with principle-driven prompting to self-instruct a pre-trained LLM. The instruction/response generation generally refers to what self-instruct and Alpaca did. The principle-driven prompting can be treated as a SFT version of Constitutional AI. Self-instructed LLaMA-65B can achieve slightly worse performance than ChatGPT, while outperforms the base LLaMA and Alpaca.

**Strengths:**

1. The principle-driven prompting is a very insightful idea to generate a large amount of fine-tuning data with minimal human annotation. The cost of labeling data for LLM alignment is a pain point in today's LLM development. The proposed method can be a worth-attempting approach for researchers with limited labeling budget.
2. The verbose version training and the discussion of verbose tax is very insightful. How to balance model's performance on specific tasks and its HHH is always an important question in the field. The authors show that simply SFT with context distillation has some limitations but can still reach better performance than non principle-driven distilled models.
3. The prompt design is very detailed and careful. Although the idea is clearly inspired by self-instruct and Constitutional AI, the details of the prompt design should be still considered as a novel contribution and helpful to the community.

**Weaknesses:**

1. The paper doesn't cover preference data generation and thus not applicable to RL tuning. A lot of OpenAI's work and talks (InstructGPT etc.) have stated the importance and necessity of RLHF. The model may finally face a performance ceiling if only using the current SFT-style self-instruct, which might be where the performance gap comes from. This is not only a con of this paper but also for those LLaMA family papers including Alpaca, Vicuna, etc.
2. The self-instruct style of data generation may lead to a narrow data distribution, e.g., the base LLM is not likely to generate a complex math problem and its answer --- and therefore the tuned model would fail to solve difficult math problems (e.g. possibly poor performance on MATH benchmark). This problem may be solved by adjusting the seed prompts and principles. But in general, how this approach generalize to different tasks is not well-discussed.

**Questions:**

line 152, 153: new instructions novel instructions. Duplicated.

**Limitations:**

In the paper, actually there is no fair comparison on fine-tuning on human labeled data v.s. fine-tuning on self-instruct generated data. Dromedary v.s. ChatGPT/InstructGPT, they have different base model. Dromedary v.s. Alpaca/Vicuna, latter's data are not human labeled. But this is a very important question for the developers in LLM alignment teams as they have to decide whether to spend money on human labeling and how much. The discussion on this topic is kind of beyond this paper, but is an important question remains in future work.

---

> ### Author Rebuttal · Authors · 2023-08-10
>
> We sincerely thank you for the thorough review and positive feedback on our work. Your questions/comments/suggestions are invaluable for the improvement of our method and the revision of the manuscript. Let us address each individual comment/question below.
>
> > On the potential performance ceiling of SFT-style self-instruct:
>
> Thanks for this insightful comment. We concur that relying solely on the current SFT-style self-instruct would not be good enough in the future, and we are actively exploring ways to push up the ceiling such as better ICL exemplars and reinforcement learning training as a part of our upcoming work.
>
> > Regarding the narrow data distribution concern with self-instruct style data generation:
>
> Great point! We envision that target-informed prompt generation, as outlined in [1], is promising for addressing this potential issue. We also recognize that tailoring the prompt generation process for distinct problem domains, such as mathematics or coding, are crucial. An alternative route could be leveraging openly collected user intentions [2] or gleaning instructions from comprehensive NLP datasets [3].  All of these directions are exciting topics for future research.
>
> > Comparing fine-tuning on human-labeled data and self-instruct generated data:
>
> Excellent point!  At the time we embarked on the Dromedary project, the availability of open-source human-labeled datasets was limited. In future work, we plan to evaluate the performance of Dromedary (or its successor, Dromedary-2) with LLaMA (or LLaMA-2) fine-tuned on human annotation datasets like Dolly [4], OpenAssistant [5] or LIMA [6] for comparison.
>
> ---
>
> [1]: Synthetic Prompting: Generating Chain-of-Thought Demonstrations for Large Language Models
>
> [2]: ShareGPT
>
> [3]: Orca: Progressive Learning from Complex Explanation Traces of GPT-4
>
> [4]: Free Dolly: Introducing the World's First Truly Open Instruction-Tuned LLM
>
> [5]: OpenAssistant Conversations -- Democratizing Large Language Model Alignment
>
> [6]: LIMA: Less Is More for Alignment

---

> > ### Comment · Reviewer_Yvp3 · 2023-08-10
> > **Acknowledging Having Read The Rebuttal**
> >
> > Thanks for the rebuttal. I've read the content and have no further questions.

---

### Official Review · Reviewer_EoHL · 2023-07-07

**Soundness:** 3 good
**Presentation:** 3 good
**Contribution:** 3 good
**Rating:** 7
**Confidence:** 4

**Summary:**

The authors study the problem of language model alignment and propose to leveraged hand-crafted prompts, principles, and examples to provide guidance, instead of relying on manually annotated human preference data. The authors make comparisons with various ai systems and the results demonstrate the effectiveness of the proposed approach.

**Strengths:**

The studied problem is important, the proposed solution is novel, and the empirical performance is good. Also, systematic analyses are provided to better understand the effectiveness of the proposed method.

**Weaknesses:**

The experiments mainly focus on demonstrating the effectiveness of the proposed algorithm. It would be better to have more analyses on the algorithm design. For example, whether the effectiveness of the algorithm has a heavy dependency on the number/quality of prompts.

**Questions:**

What are the computation requirements of the proposed algorithm?

**Limitations:**

As mentioned before, it would be better to have analyses of the performance with various number of prompts/examples.

---

> ### Author Rebuttal · Authors · 2023-08-10
>
> We sincerely thank you for the thorough review and positive feedback on our work. Your questions/comments/suggestions are invaluable for the improvement of our method and the revision of the manuscript. Let us address each individual comment/question below.
>
> > Regarding the dependence of our algorithm on the number/quality of prompts:
>
> This question is indeed perceptive. After the recent release of LLaMA-2 and its extended 4k context length, we have experimented with responses that adhered more closely to the general-specific-general style within ICL examples. The results are highly promising: our Dromedary-2 model, trained from LLaMA-2 and improved with ICL examples, has enhanced performance even without the verbose cloning phase nor inference-time few-shot examples. The results on Vicuna benchmark questions are shown below:
>
> | v.s. Vicuna-13b |  | W | T | L |
> |---|---|---|---|---|
> | Dromedary-1-65b (final) | 0-shot | 16 | 1 | 63 |
> | Dromedary-1-65b (final) | 2-shot | 19 | 8 | 53 |
> | Dromedary-2-70b (self-align-only) | 0-shot | 28 | 32 | 20 |
> | Dromedary-2-70b (self-align-only) | 2-shot | 52 | 17 | 11 |
>
> We will include these new findings in the version of our manuscript.
>
> > In relation to the computational demands of our algorithm:
>
> As outlined in our released code, a general setup would entail a minimum of 2 x 6 V100-40G GPUs for generating synthetic responses, with a more extensive requirement of 8 x 6 V100-40G GPUs for the training (LoRA fine-tuning) process.

---

### Official Review · Reviewer_VSmd · 2023-07-27

**Soundness:** 3 good
**Presentation:** 3 good
**Contribution:** 2 fair
**Rating:** 6
**Confidence:** 4

**Summary:**

Paper presents a SFT approach for instruction fine-tuning with minimal supervision.

(1) First uses self-instruct approach to augment instructions;\
(2) Using human-written rules & in-context demonstrations for thought process of response, final responses are generated by foundation models, and then distilled to model;\
(3) Further conducts context distillation to make verbose output;

Results show pretty powerful performance compared to other open-source and API models.


**Strengths:**

The proposed approach is easily comprehensible and showcases its effectiveness in producing instruction question-answer pairs to train instruction-following models with limited human supervision.
This approach holds significant importance for the open-source community as it demonstrates a strong commitment towards democratizing large language models.

**Weaknesses:**

Paper shows cost-efficient approach to create powerful instruction-tuned models but I have following minor concerns.

1. According to the evaluation results depicted in Figure 5, it becomes evident that incorporating few-shot examples is essential in achieving high-quality answers. This observation leads to the suspicion that the model must need in-context examples to generate high-quality answers since you distill outputs from in-context learning (Appendix Figure 5 also shows that zero-shot performance is worse compared to other “instruction-tuned multi-turn'' models). This observation suggests that there is still a requirement for answers generated by human writers (e.g., LIMA and LLAMA2) or responses derived from instruction-tuned models (e.g., Alpaca and Vicuna), rather than depending on outputs generated by in-context learning, despite the associated costs?

2. In-context learning pipeline described in this paper incorporates a combination of instructional guidance (rules) and few-shot examples (thought process of response). I believe instruction-tuned models may have better capability for this kind of in-context learning compared to relying on foundation models (Wei et al., 2023).

[1] LIMA: Less is more for alignment., Zhou et al., 2023\
[2] Larger language models do in-context learning differently., Wei et al., 2023


**Questions:**

(Please refer to Weakness)

---

> ### Author Rebuttal · Authors · 2023-08-10
>
> We sincerely thank you for the thorough review and positive feedback on our work. Your questions/comments/suggestions are invaluable for the improvement of our method and the revision of the manuscript. Let us address each individual comment/question below.
>
> > the need for in-context examples to produce high-quality responses:
>
> We do observe Dromedary’s performance is better with in-context examples, but it does not imply Dromedary essentially needs in-context examples.
>
> 1. We recognize that the performance of GPT-4 and similar models tend to favor lengthier responses and its inherent response patterns [1,2,3]. For instance, a typical GPT-4 response might initiate with an overview, delve into specifics, and wrap up with a summary (known as the general-specific-general style). Given that the ICL examples we used aren't particularly lengthy or aligned with this style, we leverage the two-shot examples primarily to enhance the response's length and style, as a post-training fix.
>
> 2. After the recent release of LLaMA-2 with its extended 4k context length, we have experimented with responses that adhered more closely to the general-specific-general style within ICL Self-Align examples. The results are highly promising: The Dromedary-2 model, trained from LLaMA-2 and with improved ICL exemplars, has enhanced performance even without the verbose cloning phase nor inference-time few-shot examples. The results of Vicuna benchmark questions are shown below:
>
> | v.s. Vicuna-13b |  | W | T | L |
> |---|---|---|---|---|
> | Dromedary-1-65b (final) | 0-shot | 16 | 1 | 63 |
> | Dromedary-1-65b (final) | 2-shot | 19 | 8 | 53 |
> | Dromedary-2-70b (self-align-only) | 0-shot | 28 | 32 | 20 |
> | Dromedary-2-70b (self-align-only) | 2-shot | 52 | 17 | 11 |
>
> We will include these new findings in the version of our manuscript.
>
> > the potential of instruction-tuned models for in-context learning:
>
> It's conceivable that a FLAN-style multi-task training process would boost the in-context learning and the reasoning prowess of the base LM, rendering it as an enhanced component in the Self-Align pipeline. However, the benefits from such instruction-tuned models would rely on the availability of human-annotated data while our focus in this paper is to minimize the dependency on large volumes of human annotations.
>
> ---
>
> [1] How Far Can Camels Go? Exploring the State of Instruction Tuning on Open Resources
>
> [2] The False Promise of Imitating Proprietary LLMs
>
> [3] Judging LLM-as-a-judge with MT-Bench and Chatbot Arena
>
> Once again, thank you for taking the time to review our manuscript. We will polish it further and include the discussions above. We're eager to discuss and address any concerns you may have.

---

### Decision · Program_Chairs · 2023-09-21

**Decision:**

Accept (spotlight)

**Comment:**

Meta Review for Principle-Driven Self-Alignment of Language Models from Scratch with Minimal Human Supervision

As summarized by reviewer PEmx, this work proposes a new method to generate diversified, principle-guided synthetic data from LLM itself to ease the requirement of large amounts of annotated instruction-following data for supervised fine-tuning tasks for LLMs. The generation process follows four steps. First step is to generate adversarial topic-guided instructions, the second step is to generate responses using in-context learning with pre-defined principles, and the third step is supervised fine-tuning with generated instruction-following data. Fourth step is to leverage context distillation for verbose cloning. LLaMA-65b model is fine-tuned with the data generated from the aforementioned approach. Evaluations are conducted on TruthfulQA and Bigbench-HHH Eval dataset. The fine-tuned model shows strong performance.

Most authors agree on the novelty of the method, the interesting results, and the clear presentation. A clear acceptance.